# LOOKAHEAD ANCHORING: PRESERVING CHARACTER IDENTITY IN AUDIO-DRIVEN HUMAN ANIMATION

## ABSTRACT

Audio-driven human animation models often suffer from identity drift during temporal autoregressive generation, where characters gradually lose their identity over time. One solution is to generate keyframes as intermediate temporal anchors that prevent degradation, but this requires an additional keyframe generation stage and can restrict natural motion dynamics. To address this, we propose **Lookahead Anchoring**, which leverages keyframes from future timesteps *ahead* of the current generation window, rather than within it. This transforms keyframes from fixed boundaries into directional beacons: the model continuously pursues these future anchors while responding to immediate audio cues, maintaining consistent identity through persistent guidance. This also enables self-keyframing, where the reference image serves as the lookahead target, eliminating the need for keyframe generation entirely. We find that the temporal lookahead distance naturally controls the balance between expressivity and consistency: larger distances allow for greater motion freedom, while smaller ones strengthen identity adherence. When applied to three recent human animation models, Lookahead Anchoring achieves superior lip synchronization, identity preservation, and visual quality, demonstrating improved temporal conditioning across several different architectures.

## 1 INTRODUCTION

Audio-driven human animation aims to generate realistic human videos synchronized with input audio, with widespread applications in film production, virtual assistants, and digital content creation. The advent of Diffusion Transformers (DiTs) (Peebles & Xie, 2022) has significantly advanced this field, enabling natural human video generation not only for portrait videos but also in diverse environments with complex backgrounds (Xu et al., 2024; Chen et al., 2025a). These models exhibit strong performance in capturing facial expressions, head movements, and lip synchronization across various settings. However, current DiT-based models can only handle short clips at a time, typically around 5 seconds, due to the quadratic complexity of diffusion transformer architectures.

To address this, segment-wise autoregressive generation has emerged as a promising approach, in which models synthesize video segments conditioned on preceding frames in an autoregressive fashion, in principle enabling videos of arbitrary length (Cui et al., 2024; Gan et al., 2025; Kong et al., 2025). Yet, these autoregressive strategies often suffer from character identity drift: since each new segment is conditioned on previously generated frames, small errors compound over time, causing the character's appearance to gradually deviate from the original reference.

One solution is to add reference image conditioning to anchor character features (Cui et al., 2024; Gan et al., 2025), but this creates an inherent conflict between two conditions: starting frames and reference images. The model must start from the exact final frames of the previous segment, acting as rigid constraints that cannot be violated, while simultaneously following general appearance guidance from reference images. Since reference images do not specify where character features should appear in the timeline, models often prioritize the immediate starting frame constraints, gradually losing the reference character's appearance (see Fig. 1). To mitigate identity drift, recent efforts (Bigata et al., 2025; Ji et al., 2025; Yin et al., 2023) have focused on enhancing character consistency. For instance, KeyFace (Bigata et al., 2025) generates lip-synced sparse keyframes from audio input which serve as endpoint conditions for each autoregressive segment. By enforcing these keyframes as boundary constraints along the timeline, this method successfully reduces identity drift. However,

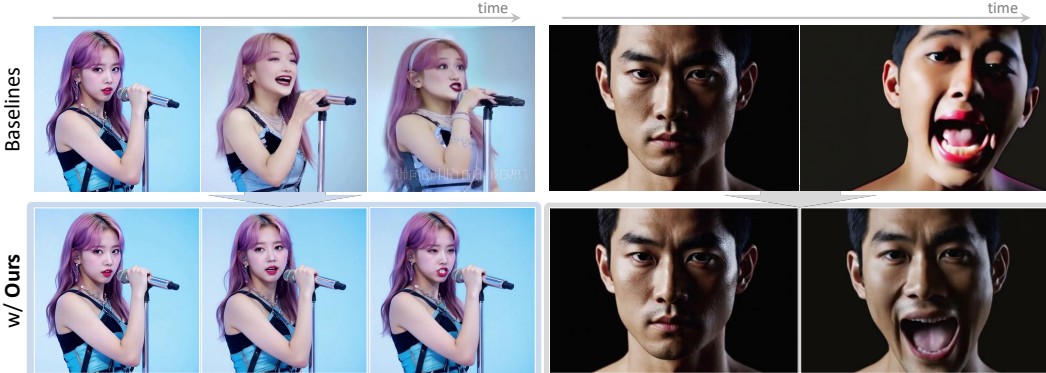

Figure 1: **Lookahead Anchoring enables robust long-form audio-driven animation.** While autoregressive generation with HunyuanAvatar (Chen et al., 2025a) (left) and OmniAvatar (Gan et al., 2025) (right) progressively loses character identity and lip sync quality, our approach maintains both throughout extended generation. We provide video results in the supplementary material.

it requires a 2-step inference of keyframe generation and subsequent interpolation and is bound by the upper limit of the quality and expressiveness of the initial keyframes.

To address the limitations introduced by keyframe interpolation, we ask: *must keyframes necessarily function as rigid boundaries for the generated segment?* To answer this, we propose **Lookahead Anchoring**, which extends the keyframe logic by leveraging keyframes that are *ahead of the segment that is being generated*. For instance, when generating the segment from 5 to 10 seconds, the keyframe is positioned at 13 seconds; for the next segment from 10 to 15 seconds, it shifts to 18 seconds, always staying 3 seconds ahead of the current generation window. This temporal separation fundamentally changes the keyframe's role: instead of imposing rigid constraints on identity, expression and pose, it provides soft directional guidance that maintains identity while enabling expressive video generation. This approach offers substantial advantages. First, the keyframe no longer needs to match the exact lip movements and expressions required by the audio at that timestamp, since it represents a distant target rather than an immediate constraint. This enables self-keyframing: directly using the reference image as a recursive anchor, removing the need for a separate keyframe generation stage. Furthermore, the temporal distance becomes a control parameter to balance between reference adherence and motion expressiveness. Our analysis reveals that longer distances increase motion dynamics while shorter distances improve facial consistency, with an optimal range that maximizes lip-sync performance.

To integrate Lookahead Anchoring with existing human animation models, we propose a tailored fine-tuning strategy, and demonstrate its effectiveness across three recent DiT-based models, proving that our findings generalize to multiple different architectures. For autoregressive generation, we find that our method achieves superior lip-sync accuracy, character consistency, and video quality compared to baselines in long video generation. Finally, we highlight the versatility of our strategy through a narrative-driven long video generation application, where it seamlessly integrates with external prompt-based image editing models (Batifol et al., 2025; Google, 2025).

## 2 RELATED WORK

**Video diffusion transformers.**  Early video diffusion models extended U-Net architectures from text-to-image generation through temporal attention modules (Guo et al., 2023; Singer et al., 2022; Chen et al., 2023; Blattmann et al., 2023). The field has since shifted to Diffusion Transformers (DiTs) (Peebles & Xie, 2022), which leverage full transformer architectures for superior scalability and spatiotemporal modeling. Recent video DiTs like Wan (Wan et al., 2025), CogVideoX (Yang et al., 2024), and HunyuanVideo (Kong et al., 2024) demonstrate improved generation quality across diverse content including human motion and dynamic scenes. These capabilities prove particularly valuable for downstream applications requiring precise temporal control.

**Audio-driven human animation.**  Early methods like SadTalker (Zhang et al., 2023; 2021) animated talking heads using 3D morphable models but struggled with expressive facial dynamics and natural lip synchronization. Diffusion models with U-Net architectures (Xu et al., 2024; Wei et al.,

2024; Chen et al., 2025b) improved temporal coherence by leveraging their pretrained video priors, though remained limited to portrait generation. DiT-based methods transformed this landscape through their superior spatiotemporal modeling capabilities, enabling diverse in-the-wild videos with full-body scenarios and complex backgrounds. Hallo3 (Cui et al., 2024) pioneered DiT-based audio-driven animation with identity reference networks and HunyuanVideo-Avatar (Chen et al., 2025a) introduced emotion control, while OmniAvatar (Gan et al., 2025) and MultiTalk (Kong et al., 2025) expanded to full-body and multi-person scenarios. Despite these advances, DiT models inherit the quadratic complexity of transformers, preventing single-pass generation of long videos.

**Long-term video generation.**  Generating long sequences with video diffusion models remains fundamentally challenging. Training-based methods like Diffusion Forcing (Chen et al., 2024) and Self Forcing (Huang et al., 2025) control per-frame noise to reduce temporal errors, yet lack identity preservation mechanisms. While FramePack (Zhang & Agrawala, 2025) prevents drift through reverse order generation, this contradicts the sequential nature of audio-driven animation. For audio-driven human animation, DiT-based approaches including OmniAvatar (Gan et al., 2025) and Hallo3 (Xu et al., 2024) employ temporal autoregression with reference conditioning, but identity degradation persists beyond tens of seconds due to error accumulation. Sonic (Ji et al., 2025) employs sliding windows at inference but, without training integration, achieves limited improvement in long term consistency. KeyFace (Bigata et al., 2025) successfully reduces drift through keyframe generation and interpolation, but requires an additional model and rigidly constrains motion to predetermined keyframes, limiting expressiveness. While we share KeyFace's temporal anchoring insight, our approach eliminates keyframe generation overhead and transforms hard constraints into flexible guidance, enabling arbitrary-length generation with preserved expressiveness. OmniHuman-1.5 (Jiang et al., 2025), a concurrent work, achieves long-term generation through a similar temporal anchoring approach, which we discuss further in Appendix E.

## 3 METHOD

We address the task of generating arbitrarily long, audio-driven human animation videos while maintaining character identity and video quality. Given an audio sequence $\mathbf{a}$ and a reference image $I_{\text{ref}}$, the goal is to generate a video that maintains consistent identity with $I_{\text{ref}}$ while exhibiting natural motion synchronized to $\mathbf{a}$. In this section, we first establish the limitations of existing keyframe-based methods (Sec. 3.1), then introduce our concept of Lookahead Anchoring (Sec. 3.2), and finally detail the integration of this approach into video DiTs (Sec. 3.3).

### 3.1 PRELIMINARIES AND MOTIVATION

**Temporal autoregressive video generation.**  Conventional temporal autoregressive approaches tackle this by dividing the video into segments of length $L$, as in Gan et al. (2025); Cui et al. (2024). For the $i$-th segment $V_i$ spanning frames $[iL, (i+1)L)$, the generation process $G_{\text{auto}}(\cdot)$ is conditioned on the corresponding audio segment $\mathbf{a}_{iL:(i+1)L}$ and the final frames of the previous segment $V_{i-1}^{\text{end}}$:

$$V_i = G_{\text{auto}}(\mathbf{a}_{iL:(i+1)L}, V_{i-1}^{\text{end}}, I_{\text{ref}}). \tag{1}$$

While this enables theoretically unlimited video length, errors in $V_{i-1}^{\text{end}}$ accumulate across segments, causing progressive identity drift and quality degradation.

To preserve character identity, several methods implement the reference conditioning for $I_{\text{ref}}$. For instance, Hallo3 (Xu et al., 2024) employs feature concatenation through ReferenceNet (Hu, 2024), while OmniAvatar (Gan et al., 2025) uses channel concatenation for reference injection. However, we empirically find that none of these strategies are sufficient for maintaining long-term identity consistency, as shown in Fig. 1 and 4.

**Keyframe-based anchoring.**  Recent keyframe-based methods such as KeyFace (Bigata et al., 2025) address identity drift by introducing temporal anchors throughout generation. Specifically, these approaches employ a two-stage framework. First, a specialized audio-conditional model $G_{\text{key}}$ generates sparse keyframes $\mathcal{K} = \{k_{L-1}, k_{2L-1}, ...\}$ where each $k_t$ is a single frame synchronized with audio at time $t$:

$$k_t = G_{\text{key}}(\mathbf{a}_{t-\epsilon:t+\epsilon}, I_{\text{ref}}), \tag{2}$$

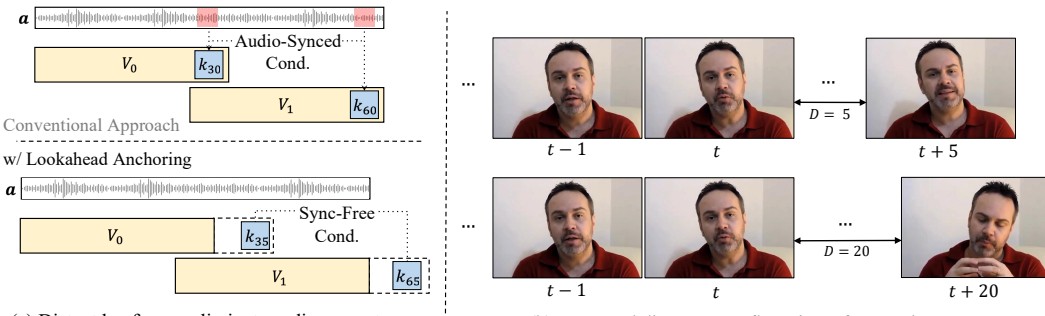

(a) Distant keyframes eliminate audio-sync stage.

(b) Temporal distance $D$ reflects inter-frame relevance.

Figure 2: **Motivation.** (a) We depart from the convention of using conditional keyframes as generation window endpoints. Instead, we reposition keyframes as temporally distant anchors beyond the window, decoupling them from the actual generated sequence. This eliminates constraints such as audio synchronization requirements while enabling flexible conditioning. (b) Models naturally learn that longer temporal distances allow for greater scene variation. We exploit this prior strategically: distant keyframes provide high-level guidance without imposing strict physical constraints, enabling diverse yet coherent generation.

where $\epsilon$ is the context window size. Then, each video segment $V_i$ is generated with keyframes serving as boundary constraints:

$$V_i = G_{\text{interp}}(\mathbf{a}_{iL:(i+1)L}, k_{iL}, k_{(i+1)L-1}), \tag{3}$$

where $G_{\text{interp}}(\cdot)$ is a generative video interpolation model, with $k_{iL}$ and $k_{(i+1)L-1}$ serving as boundary constraints for segment $V_i$.

This two-stage approach successfully reduces drift by providing regular identity anchors, but introduces its own limitations: it requires training an additional keyframe generation model, increasing overall complexity; it constrains the model to produce exact poses at predetermined timestamps, which can suppress natural temporal dynamics and reduce flexibility in motion generation; and it limits the visual quality of the final video, since it is ultimately upper-bounded by the quality of the generated keyframes themselves.

### 3.2 LOOKAHEAD ANCHORING

To overcome these limitations while retaining the identity-anchoring benefits of keyframes, we propose **Lookahead Anchoring**. Our key idea is simple: instead of forcing the model to reach specific keyframes at segment boundaries, we position them perpetually ahead, always visible but never reached.

Formally, when generating segment $V_i$, our model $G_{\text{LA}}(\cdot)$ conditions on a keyframe positioned $D$ frames beyond the segment endpoint, where this lookahead distance $D \in \mathbb{N}, D > 0$ becomes a control parameter:

$$V_i = G_{\text{LA}}(a_{iL:(i+1)L}, V_{i-1}^{\text{end}}, k_{(i+1)L-1+D}), \tag{4}$$

where $V_{i-1}^{\text{end}}$ denotes starting frames to ensure temporal continuity. This modification changes how the model interprets the keyframe. Instead of being rigidly bound to match the keyframe at the segment's edge, the model is encouraged to generate expressive motion that progressively moves towards the target appearance. It is perpetually "*chasing*" the keyframe, which maintains a consistent distance from each generation window throughout the autoregressive process. It ensures that the model never directly reaches the keyframe target while continuously receiving its guidance signal.

**Lookahead distance as degree of keyframe relevance.** Following this intuition, the lookahead temporal distance $D$ can be seen as a parameter that controls the keyframe's influence. A shorter distance grounds the model more firmly in the keyframe, whereas a longer one affords greater motion freedom, suggesting a natural way to balance identity adherence and expressiveness. We empirically verify this relationship in Sec. 4.3, where Fig. 6 demonstrates that increasing temporal distance systematically trades facial consistency for enhanced motion dynamics.

**Sync-free keyframes.** A fundamental limitation of traditional keyframe methods is the tight coupling of identity and motion cues within a single audio-synchronized frame. Positioning the keyframe in the distant future decouples identity and motion: the keyframe anchors identity, while the current audio drives motion.

This temporal separation eliminates the need for an audio-synchronized keyframe, as precise audio-pose alignment at that distant timestamp is no longer necessary. Consequently, any image can serve as a directional identity anchor during inference. We can formally replace the keyframe with a condition image $I_{\text{trg}}$:

$$k_{(i+1)L-1+D} \leftarrow I_{\text{trg}}. \tag{5}$$

For instance, we can directly leverage prompt-based image editing models (*e.g.*, FLUX (Batifol et al., 2025), Nano Banana (Google, 2025)) to generate diverse $I_{\text{trg}}$ representing different expressions, poses, or contexts, enriching the final video without training specialized audio-conditional models (see Fig. 7)

**Self-keyframing.** Crucially, this decoupling also enables a novel and effective strategy we call Self-keyframing: leveraging the reference image $I_{\text{ref}}$ as the perpetual distant target, i.e., $I_{\text{trg}} := I_{\text{ref}}$. This further simplifies the framework, by *eliminating the keyframe generation stage entirely* while maintaining consistent identity guidance. Unlike conventional reference conditioning that injects static features, our approach assigns $I_{\text{ref}}$, a specific future temporal position. The model interprets this timestamped reference as a future target, creating persistent identity pull throughout generation.

### 3.3 Leveraging Lookahead Anchoring for Audio-Conditional Video DiTs

Modern video DiTs generally process noisy video latents through multiple layers of full 3D attention, progressively denoising them across diffusion timesteps. A VAE encoder produces spatiotemporally compressed latent representations, which are then patchified and flattened into token sequences. Each token receives a positional embedding $p_{x,y,t} = p_x \oplus p_y \oplus p_t$, where $\oplus$ denotes concatenation, encoding its spatial $(x, y)$ and temporal $(t)$ position. For a sequence of $n$ latent frames, we denote the temporal embedding at position $\ell$ as $p_t[\ell]$ where $\ell \in \{0, 1, ..., n-1\}$. These $n$ latent frames correspond to $L$ original video frames after temporal compression with ratio $r = L/n$. We focus on manipulating temporal embeddings to encode distant inter-frame relationships.

**Distant keyframe conditioning.** We realize this by assigning the keyframe tokens to positions corresponding to distant future timestamps in the DiT input sequence. During training, given a latent sequence $\mathbf{z} = \{z_0, ..., z_{n-1}\}$ where each $z_i$ denotes the set of spatial tokens of frame $i$, we append a distant clean latent $z_{n-1+d}$ as a condition to form $\mathbf{z}' = \{z_0, ..., z_{n-1}, z_{n-1+d}\}$, where $d = \lfloor D/r \rfloor$ is the temporal distance in latent frames. As the conditioning latent remains clean throughout the denoising process, we handle this asymmetry with a projection layer $\phi$ that maps clean conditioning latents into the space of noisy generation tokens. The appended latent receives temporal embedding $p_t[n-1+d]$, signaling its future position.

At inference, we replace $z_{n-1+d}$ with $z_{\text{trg}}$, a latent of the encoded conditional image $I_{\text{trg}}$ while maintaining the distant time positional embedding:

$$\mathbf{z}' = \texttt{concat}\big(\{z_0, ..., z_{n-1}\}, z_{\text{trg}}\big) \quad \text{with} \quad \mathbf{p}'_t = \big\{p_t[0], ..., p_t[n-1], p_t[n-1+d]\big\}, \tag{6}$$

where $\texttt{concat}(\cdot)$ denotes concatenation. This design enables the model to leverage temporal distance as a control signal while maintaining clean identity guidance throughout the denoising process.

**Do video DiTs understand distant frames?** Our approach presumes that video DiTs can meaningfully process frames at non-consecutive temporal positions. To validate this assumption, we investigate whether pretrained video DiTs generalize to distant temporal relationships. We conduct a simple experiment: using a pretrained audio-conditional video DiT (Gan et al., 2025), we generate two-frame videos, conditioning on the first frame while manipulating the temporal positional embedding of the second frame to simulate an artificially large time gap (detailed in Appendix B).

Fig. 3 demonstrates that the pretrained model exhibits an adaptive behavior: increased temporal distances yield larger motion magnitudes, suggesting the model has an intrinsic understanding of temporal dynamics. However, without explicit training on distant frame pairs, the generation quality deteriorates under distant conditioning, producing artifacts and inconsistent identity. This finding validates our hypothesis that video DiTs contain exploitable temporal structure, while simultaneously motivating a fine-tuning strategy to refine this raw capability into a precise control mechanism.

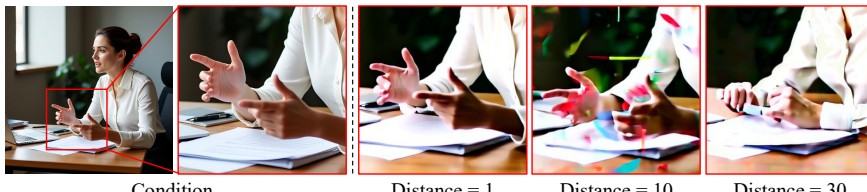

| Condition | | Distance = 1 | Distance = 10 | Distance = 30 |

Figure 3: **Exploration of distant frame relationships in a pretrained video DiT (Gan et al., 2025).** Given a conditional frame, we generate separate two-frame videos with artificially increased temporal gaps. Testing beyond the training distribution naturally degrades visual quality but reveals adaptive motion behavior. We propose fine-tuning to harness this observed temporal structure.

**Fine-tuning strategy.** A naive approach of fine-tuning exclusively with distant keyframes would force the model to abruptly abandon its learned temporal dynamics. Instead, we sample keyframes from a continuous range of temporal positions during training.

Given a training video sequence, we sample the keyframe position $\ell$ from a range that spans both inside and outside the current generation window:

$$\ell \sim \mathcal{U}[0, n-1+d_{\max}], \quad \text{where} \ \ d_{\max} > 0, \tag{7}$$

where $d_{\max}$ denotes the maximum temporal distance in latent frames during training. When $\ell$ falls within the generation window ($\ell < n$), the model learns to produce accurate reconstructions at the specified position. When $\ell$ exceeds the window boundary ($\ell \geq n$), the model learns to modulate the keyframe's influence based on its temporal distance, thereby instantiating the desired soft guidance mechanism. This soft exposure allows the model to learn a smooth and generalizable function of how the keyframe's influence attenuates with distance, which we explore further in Sec. 4.3.

## 4 EXPERIMENTS

### 4.1 EXPERIMENTAL SETUP

To demonstrate its generalizability, we integrate our approach into three audio-driven human animation DiTs (base models): Hallo3 (Cui et al., 2024), HunyuanVideo-Avatar (Chen et al., 2025a), and OmniAvatar (Gan et al., 2025). We fine-tune on the Hallo3 training dataset and 160 hours of collected portrait video data. All baseline DiT models are fine-tuned on the same data to ensure a fair comparison. We evaluate on HDTF (Zhang et al., 2021) and AVSpeech (Ephrat et al., 2018) test splits, sampling videos exceeding 30 seconds (average length: 48s), which require 9-15 iterative segment generations. HDTF provides portrait evaluation (shoulder-to-face), while AVSpeech enables in-the-wild assessment with complex backgrounds and varied body compositions. For fine-tuning, we follow the original training configurations of each baseline DiT, including loss functions and learning rates. Fine-tuning details for each baseline model are provided in Appendix B. Code and weights will be publicly released.

### 4.2 ADAPTING STATE-OF-THE-ART MODELS WITH LOOKAHEAD ANCHORING

**Quantitative comparisons.** Tab. 1 and 2 present quantitative evaluations on HDTF (Zhang et al., 2021) and AVSpeech (Ephrat et al., 2018), comparing baselines with our approach. Following established protocols (Zhang et al., 2023; Xu et al., 2024), we report SyncNet (Prajwal et al., 2020) distance and confidence for lip synchronization quality. Face and subject consistency are measured via cosine similarity of ArcFace (Deng et al., 2019) and DINO (Zhang et al., 2022) features, respectively, following VBench (Huang et al., 2024). We additionally report FID (Heusel et al., 2017), FVD (Unterthiner et al., 2018), and motion smoothness (MS) (Huang et al., 2024) to assess overall generation quality. To analyze temporal stability, we compute FID scores using 1-second sliding windows normalized by the initial window in Fig. 5. While baseline methods show progressive degradation over time, our approach maintains consistent quality, with the most substantial improvements observed in OmniAvatar, followed by HunyuanAvatar and Hallo3. User studies in Tab. 3 demonstrate superior preference for our approach across all baselines in lip synchronization, character consistency, and overall quality, among 34 participants. Detailed user study protocols are described in Appendix C.

Table 1: **Quantitative results on HDTF (Zhang et al., 2021) in temporal segment-wise autoregressive scheme.** We compare temporal autoregressive long video generation of DiT-based models with and without our proposed approach. Additionally, we provide comparisons with five GAN- and Diffusion U-Net-based models. For a fair comparison, all three DiT-based methods are fine-tuned on the same training data used in our approach. * HunyuanAvatar is modified to take 5 starting frames to compare in the temporal autoregressive generation scheme.

| Models | Sync-D ↓ | Sync-C ↑ | Face-Con. ↑ | Subj-Con. ↑ | FID ↓ | FVD ↓ | MS ↑ |
|---|---|---|---|---|---|---|---|
| SadTalker (Zhang et al., 2023) | 7.64 | 7.59 | 0.9326 | 0.9934 | 86.20 | 423.18 | 0.9956 |
| Hallo (Xu et al., 2024) | 7.96 | 7.39 | 0.9245 | 0.9893 | 37.27 | 243.11 | 0.9944 |
| AniPortrait (Wei et al., 2024) | 11.60 | 3.52 | 0.9235 | 0.9903 | 37.20 | 278.09 | 0.9954 |
| EchoMimic (Chen et al., 2025b) | 8.66 | 7.31 | 0.9254 | 0.9920 | 44.56 | 444.14 | 0.9949 |
| KeyFace (Bigata et al., 2025) | 10.28 | 4.91 | 0.9158 | 0.9902 | 17.01 | 157.33 | 0.9952 |
| Hallo3 (Cui et al., 2024) | 7.89 | 7.79 | 0.8628 | 0.9774 | 21.61 | 180.30 | 0.9939 |
| **+ LA (Ours)** | **7.53** | **8.22** | **0.9267** | **0.9843** | **12.44** | **142.97** | **0.9939** |
| HunyuanAvatar* (Chen et al., 2025a) | 7.77 | 8.27 | 0.6109 | 0.9470 | 37.84 | 501.80 | 0.9925 |
| **+ LA (Ours)** | **7.70** | **8.27** | **0.9135** | **0.9870** | **15.98** | **182.23** | **0.9946** |
| OmniAvatar Gan et al. (2025) | 8.48 | 7.47 | 0.7904 | 0.9568 | 35.26 | 467.90 | **0.9953** |
| **+ LA (Ours)** | **7.19** | **9.28** | **0.8628** | **0.9686** | **24.09** | **302.71** | 0.9923 |

Table 2: **Quantitative results on AVSpeech (Ephrat et al., 2018) in temporal segment-wise autoregressive scheme.** We compare temporal autoregressive long video generation of DiT-based models with and without our proposed approach.

| Models | Sync-D ↓ | Sync-C ↑ | Face-Con. ↑ | Subj-Con. ↑ | FID ↓ | FVD ↓ | MS ↑ |
|---|---|---|---|---|---|---|---|
| Hallo3 Xu et al. (2024) | 9.87 | 4.36 | 0.7422 | 0.9519 | 44.67 | 357.49 | **0.9950** |
| **+ LA (Ours)** | **8.40** | **6.03** | **0.8355** | **0.9734** | **24.96** | **351.90** | 0.9940 |
| HunyuanAvatar* Chen et al. (2025a) | 8.45 | 6.49 | 0.5282 | 0.9415 | 59.15 | 455.52 | 0.9954 |
| **+ LA (Ours)** | **8.27** | **6.55** | **0.8943** | **0.9879** | **22.89** | **271.15** | **0.9963** |
| OmniAvatar Gan et al. (2025) | 8.51 | 6.41 | 0.6378 | 0.9268 | 81.15 | 601.12 | **0.9966** |
| **+ LA (Ours)** | **8.10** | **7.36** | **0.8415** | **0.9670** | **35.83** | **510.93** | 0.9946 |

**Qualitative comparisons.** Fig. 4 presents qualitative results on AVSpeech (Ephrat et al., 2018) and HDTF (Zhang et al., 2021) under a temporal autoregressive scheme, where the last frames from previous segments initialize subsequent generation. We evaluate three audio-conditional DiTs with and without our proposed approach. While baseline methods exhibit gradual identity drift and diverge from reference image details as generation progresses, our method maintains consistent character identity and superior visual quality throughout. More qualitative results are provided in Fig. 8, 9 and 10 in the appendix. Video results in the supplementary material further demonstrate these improvements.

**Comparisons with other long video approaches.** Tab. 4 compares our method with other long video generation strategies on AVSpeech. Based on HunyuanAvatar, we evaluate a time-aware position shifting technique proposed in Sonic (Ji et al., 2025), a KeyFace-style approach that first generates sparse audio-synchronized keyframes and then fills intermediate frames through video interpolation, and a past-time conditioning variant that uses anchor frames located 8 frames before the generation window. Our lookahead anchoring achieves the best lip synchronization performance and facial consistency while maintaining high motion smoothness and eliminating the need for additional architectural complexity or separate generation stages.

### 4.3 ANALYSIS

**Temporal distance.** We analyze the impact of temporal distance between keyframes and generation windows in Fig. 6. We vary the distance from 4 to 80 frames and measure its effect on facial consistency, dynamic degree (quantified as the variance of facial landmarks across frames (Ma et al., 2023)), and lip synchronization quality. We find that increasing the distance yields higher dynamic degree while shorter distances strengthen identity preservation. Remarkably, lip synchronization peaks arounds 12 frames, revealing an optimal zone that maximizes both expressiveness and character consistency.

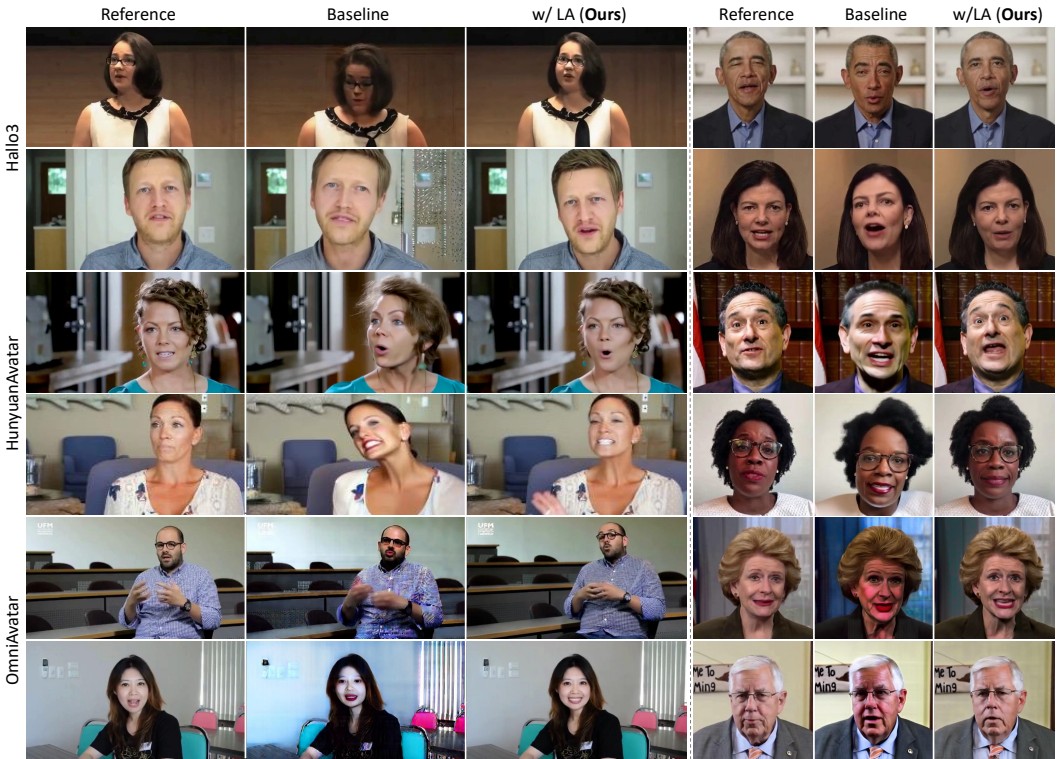

Figure 4: **Qualitative results.** We compare our method with three audio-conditioned DiT baselines under the temporal sement-wise autoregressive framework on AVSpeech (Ephrat et al., 2018) and HDTF (Zhang et al., 2021), presenting mid-sequence frames to demonstrate generation quality. Video results are available in the supplementary material.

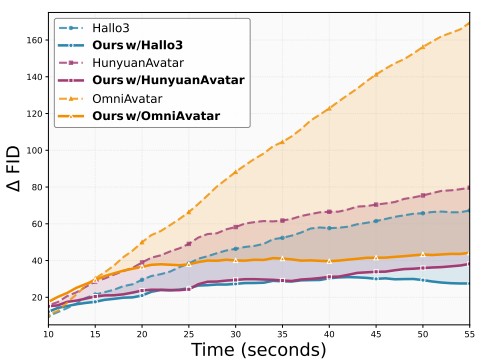

Figure 5: **Performance over time.** We report FID computed with 1-second sliding windows, normalized relative to the first window.

**Training strategy.** We compare two training approaches in Tab. 5: fixed anchoring, which fixes the keyframes' position at $\ell = n + d$, and our flexible anchoring, which samples keyframe positions from both inside and outside the generation window. The latter forces the model to learn how the keyframe's influence varies with temporal distance, encouraging further generalization. In practice, we find that flexible anchoring achieves superior lip synchronization and facial consistency.

Table 3: **User study.**

| Models | Lip-Sync. | Character-Con. | Overall Quality |
|---|---|---|---|
| Hallo3 | 21.6% | 17.6% | 25.5% |
| + LA (Ours) | **79.4%** | **82.4%** | **74.5%** |
| HunyuanAvatar* | 21.6% | 10.9% | 15.7% |
| + LA (Ours) | **79.4%** | **89.2%** | **84.3%** |
| OmniAvatar | 46.1% | 29.4% | 36.3% |
| + LA (Ours) | **53.9%** | **70.6%** | **63.7%** |

Table 4: **Comparison with other long-term video generation approaches.**

| Approach | Sync-D | Sync-C | Face-Con | MS |
|---|---|---|---|---|
| Baseline | 8.45 | 6.49 | 0.5282 | 0.9954 |
| w/ Sonic | 8.58 | 6.44 | 0.8900 | 0.9954 |
| w/ Keyframe Gen. | 8.33 | 6.53 | 0.8685 | **0.9964** |
| w/ Past-time Cond. | 8.55 | 6.36 | 0.8753 | 0.9946 |
| w/ Ours | **8.27** | **6.55** | **0.8943** | 0.9963 |

Table 5: **Ablation on anchoring strategies.**

| Approach | Sync-D | Sync-C | Face-Con |
|---|---|---|---|
| Train w/ fixed anchor | 8.50 | 6.45 | 0.8859 |
| Train w/ flexible anchor | **8.27** | **6.55** | **0.8943** |
| No time P.E. | 11.17 | 2.78 | 0.7251 |
| Learnable time P.E. | 8.68 | 6.16 | 0.6068 |
| Distant time P.E. | **8.27** | **6.55** | **0.8943** |

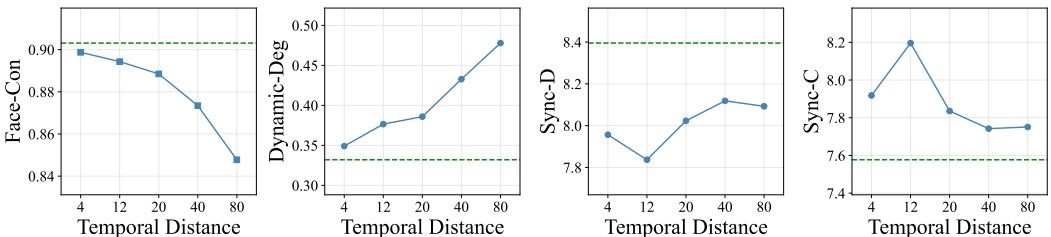

Figure 6: **Effect of temporal distance between conditional keyframes and generation windows.** As we increase the temporal distance, our approach (blue line) yields increased dynamicity over the baseline (green line), at the expense of facial consistency. On the other hand, lip synchronization is noticeably improved when the temporal distance is smaller ($\leq 12$), but degrades sharply when it is increased further.

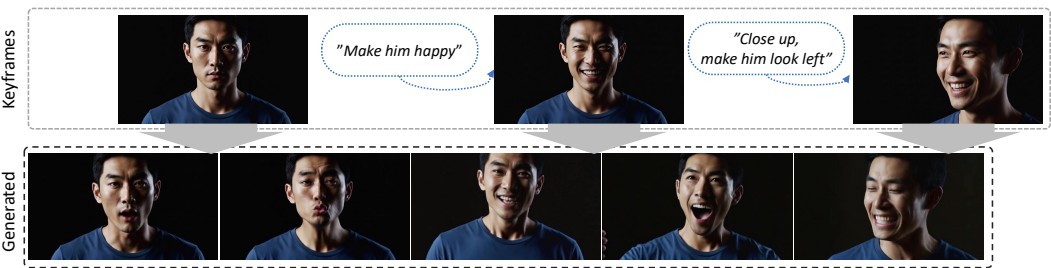

Figure 7: **Narrative-driven generation.** We edit a reference image using text prompts to create keyframes with different states, then position them as lookahead anchors during autoregressive generation to guide narrative transitions while maintaining audio synchronization. More results can be found in Fig. 11.

**Ablation on time positional embeddings.** We ablate different temporal embedding strategies for the conditional keyframe in Tab. 5. We compare our approach of assigning distant future temporal embeddings against two alternatives: learnable embeddings optimized during training, and zero embeddings without positional information. Our distant future embedding achieves the best performance, as it explicitly encodes the temporal relationship between the generation window and the conditional keyframe.

## 4.4 NARRATIVE-DRIVEN GENERATION WITH EXTERNAL IMAGE MODELS

Our lookahead anchoring allows the conditioning keyframe to be any image with consistent character identity, as it serves as a distant target rather than an immediate constraint requiring audio synchronization. Leveraging this flexibility, we generate narrative-driven videos by using text-based image editing models to create diverse keyframes representing different emotional states or contexts from a single reference image. By conditioning on these narrative keyframes at appropriate intervals, we produce dynamic videos that follow desired storylines while maintaining natural audio synchronization, as demonstrated in Fig. 7 using Nano Banana (Google, 2025). Crucially, the model does not replicate these keyframes exactly but rather converges toward similar states (*e.g.*, emotions, expressions) through soft guidance, preserving natural, audio-synchronized motion.

## 5 CONCLUSION

We introduced Lookahead Anchoring, a strategy that achieves robust identity preservation in long video generation by positioning keyframes at temporally distant future timesteps. This approach eliminates the need for specialized keyframe generation models, while providing intuitive control over the identity-expressiveness trade-off through temporal distance. The consistent improvements across three architectures demonstrate that rethinking the fundamental role of keyframes, from rigid constraints to directional guidance, offers a practical path toward high-quality, arbitrarily long audio-driven human animations.

ETHICS STATEMENT

Our method generates highly realistic long-form human animation videos, offering significant benefits for content creation, education, and accessibility applications. We acknowledge the potential risks of misuse, including unauthorized impersonation and misinformation. By releasing our work as open-source, we will aim to promote transparency and enable the development of detection mechanisms while advancing scientific understanding.

REPRODUCIBILITY STATEMENT

For reproducibility, we provide detailed implementation information in Section B in the appendix, and will release our code and model checkpoints.

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

# APPENDIX

## A  MORE QUALITATIVE RESULTS

We provide additional qualitative results to further demonstrate the effectiveness of our approach across various scenarios and applications.

**Diverse characters and environments.**  Fig.8 presents qualitative comparisons across a wide range of subjects, backgrounds, and lighting conditions, based on AI-generated image inputs. Each row shows the progression of generated frames at different timestamps, comparing baseline methods with our approach. These results demonstrate that our method consistently preserves character identity and maintains visual quality.

**Extended qualitative comparisons on AVSpeech.**  Fig. 9 and 10 provide extensive qualitative comparisons on the AVSpeech dataset (Ephrat et al., 2018). For each example, we show baseline models in the top rows and our approach in the bottom rows, presenting frames throughout the generation sequence. These comparisons reveal that while baseline methods suffer from progressive identity drift, particularly visible in facial features and hair details after multiple autoregressive steps, our method maintains consistent character appearance and preserves fine facial details throughout the entire sequence.

**Narrative-driven genration application.**  Fig. 11 showcases the versatility of our approach through narrative-driven video generation. By leveraging an external text-based image editing model (Google, 2025) to create keyframes representing different emotional states or contexts, we demonstrate how our method can generate videos that follow desired storylines while maintaining natural audio synchronization. The figure illustrates smooth transitions between different narrative states, from neutral expressions to various emotional responses, all while preserving character identity and lip synchronization quality.

## B  IMPLEMENTATION DETAILS

**Experimental setup for the pilot study in Fig. 3.**  We conduct our pilot study using the OmniAvatar (Gan et al., 2025) model to explore how pretrained video DiTs handle temporally distant frames. The model employs a video VAE encoder that compresses $N$ input frames into $(N-1)/4+1$ latent frames through asymmetric temporal compression: the first frame maintains a 1:1 correspondence in latent space, while subsequent frames undergo 4:1 temporal compression.

To generate two frame videos with controlled temporal gaps with a distance of $k$, we position the conditioning frame at temporal index $\ell = k$ in the latent space while placing the frame to be generated at index $\ell = 0$ (initialized with noise). For instance, when setting the latent distance to 3, we assign the conditioning frame positional embedding $\mathbf{p}_t[3]$ and generate the target frame at position $\mathbf{p}_t[0]$. This configuration ensures the generated content corresponds to exactly one frame in pixel space after VAE decoding. By maintaining the generated frame at position 0, we preserve the direct correspondence between latent and pixel space representations, enabling precise evaluation of motion magnitude as a function of simulated temporal distance.

### B.1  APPLYING OUR APPROACH TO EACH ANIMATION MODEL

We detail the specific implementation of our approach for each of the three audio-conditional DiT models: Hallo3 (Cui et al., 2024), HunyuanVideo-Avatar (Chen et al., 2025a), and OmniAvatar (Gan et al., 2025), which are built upon CogVideoX (Yang et al., 2024), HunyuanVideo (Kong et al., 2024), and Wan2.1 (Wan et al., 2025), respectively. We train each model on 4 NVIDIA H200 GPUs with a batch size of 8, which takes approximately 2 days.

**HunyuanVideo-Avatar (Chen et al., 2025a) and HunyuanVideo (Kong et al., 2024).** HunyuanVideo-Avatar extends the HunyuanVideo model by incorporating reference images through three pathways: LLaVA (Liu et al., 2023) embeddings, feature addition via projection layers, and

token concatenation along the sequence dimension. To integrate our approach, we modify the temporal positional embeddings of the concatenated reference tokens, assigning them positions beyond the generation window to encode their temporal distance. Specifically, we introduce a seperate projection layer for the distant conditional frame, initialized from the weights of the existing projection layer for denoising tokens. Since HunyuanVideo-Avatar employs the sliding window mechanism proposed in Sonic for long video scenario, we adapt it for temporal autoregressive generation by incorporating 5 starting frames as boundary conditions, by concatenating their tokens to the input sequence. Each generation chunk produces 101 frames conditioned on the preceding 5 frames and the reference frame. All other training settings follow the standard procedures of HunyuanVideo and HunyuanVideo-Avatar. In our experiments, we compare our method against the baseline that uses 5 starting frames with LLaVA embeddings and feature addition for reference conditioning, evaluating the impact of adding temporally distant keyframe conditioning.

**OmniAvatar (Gan et al., 2025) and Wan2.1 (Wan et al., 2025).** OmniAvatar builds upon Wan2.1, performing audio-conditional human animation by fine-tuning only the audio injection module and parameters for Low-Rank Adaptation (LoRA). The reference image is injected through channel-wise concatenation with the DiT input tokens, while starting frames are conditioned via token replacement (replacing the initial portion of noisy video tokens with clean starting frame tokens) to enable temporal autoregressive generation. To integrate our approach, we adopt a similar strategy to HunyuanVideo-Avatar: distant keyframes are conditioned through token concatenation along the sequence dimension with a separate projection layer, while their temporal positional embeddings are set to positions beyond the generation window. Following OmniAvatar's training protocol, we fine-tune only the LoRA parameters while keeping all other parameters frozen. All remaining settings follow the standard OmniAvatar training procedures.

**Hallo3 (Cui et al., 2024) and CogVideoX (Yang et al., 2024).** Hallo3 leverages CogVideoX as its base model, implementing reference image conditioning through a transformer-based reference network and face embeddings from a face encoder. Building upon Hallo3's architecture, we implement our distant keyframe conditioning through the existing reference network. Unlike Hallo3's original conditioning scheme, we apply temporal positional embeddings to the reference network input using the same encoding format as the DiT's positional encoding, enabling the network to perceive temporal distance between the current generation window and the distant keyframe.

## C USER STUDY PROTOCOL

We conducted a comprehensive user study with 34 participants to evaluate the perceptual quality improvements when our approach is applied to existing baseline methods. Each participant evaluated 9 randomly selected video pairs from a pool of 69 pre-generated pairs, resulting in a total of 306 paired comparisons. The study design ensured balanced evaluation across different baseline models. For each participant, the 9 video pairs were distributed as follows: 3 pairs comparing HunyuanVideoAvatar baseline with our method applied on top of it, 3 pairs for OmniAvatar comparisons, and 3 pairs for Hallo comparisons. For each video pair, participants were asked to indicate their preference based on four distinct criteria:

- **Character Consistency:** Which model shows better character consistency (the person in the video is similar to the input image)?

- **Overall Quality:** Which model shows better overall quality results?

- **Expression Realism:** Which model shows more natural and realistic expressions and motion?

- **Lip Sync Performance:** Which model shows better lip sync performance?

The presentation order of our method and the baseline was randomized and anonymized for each comparison. Participants were not informed which video corresponded to which method. An example of the user interface presented to participants is shown in Fig. 12. The full results of the user study are summarized in Tab. 6.

| Models | Lip-Sync Performance | Character Consistency | Expression Realism | Overall Quality |
|---|---|---|---|---|
| Hallo3 | 21.6% | 17.6% | 21.6% | 25.5% |
| **+ LA (Ours)** | 79.4% | 82.4% | 78.4% | 74.5% |
| HunyuanAvatar* | 21.6% | 10.9% | 26.5% | 15.7% |
| **+ LA (Ours)** | 79.4% | 89.2% | 73.5% | 84.3% |
| OmniAvatar | 46.1% | 29.4% | 39.2% | 36.3% |
| **+ LA (Ours)** | 53.9% | 70.6% | 60.8% | 63.7% |

Table 6: **User study result.**

## D  LIMITATIONS AND FUTURE WORK

Our approach is designed to leverage pretrained video priors from base DiT models through minimal architectural modifications. Consequently, when certain base models exhibit specific limitations, such as challenges in generating precise hand gestures or complex body movements, our framework inherently preserves these characteristics. Additionally, when text prompts specify dramatic scene transitions that significantly deviate from the reference image context (*e.g.*, transitioning from an indoor setting to outdoor environments), the model may struggle to synthesize plausible transitions. While our narrative-driven generation demonstrates that strategically placed keyframe anchors can facilitate such transitions, exploring fully dynamic scene changes with extreme background variations remains an interesting direction for future research.

## E  DISCUSSION ON CONCURRENT WORK

We discuss the relationship between our work and OmniHuman-1.5 (Jiang et al., 2025), a concurrent work that also addresses long-term audio-driven human animation through future frame conditioning.

**Shared insights.**  Both methods identify temporal anchoring via future frames as an effective strategy for mitigating identity drift in long video generation. This finding validates the importance of bidirectional temporal conditioning for maintaining consistency in long sequences.

**Methodological distinctions.**  The approaches differ primarily in their training philosophy and implementation.

**Training strategy:** OmniHuman-1.5 treats the reference image as potentially constraining motion dynamics and excludes it during training, learning solely from sequential video frames. The reference image at future position is introduced only at inference time. Our method explicitly incorporates future-sampled conditions during training, enabling the model to learn temporal distance relationships and adaptively modulate the influence of anchoring frames based on their temporal proximity.

**Temporal modeling:** Our approach introduces learnable temporal conditioning that allow the model to understand and utilize the temporal distance between the current frame and the lookahead anchor, enabling flexible control over the strength of conditioning based on temporal gaps, whereas OmniHuman-1.5 employs a fixed injection strategy at inference.

These approaches demonstrate the robustness of temporal anchoring as a principle for long-term video generation while highlighting different paths toward its implementation.

## F  THE USE OF LARGE LANGUAGE MODELS

The application of Large Language Models in writing this manuscript was confined to proofreading for grammatical accuracy and enhancing clarity by refining phrasing.

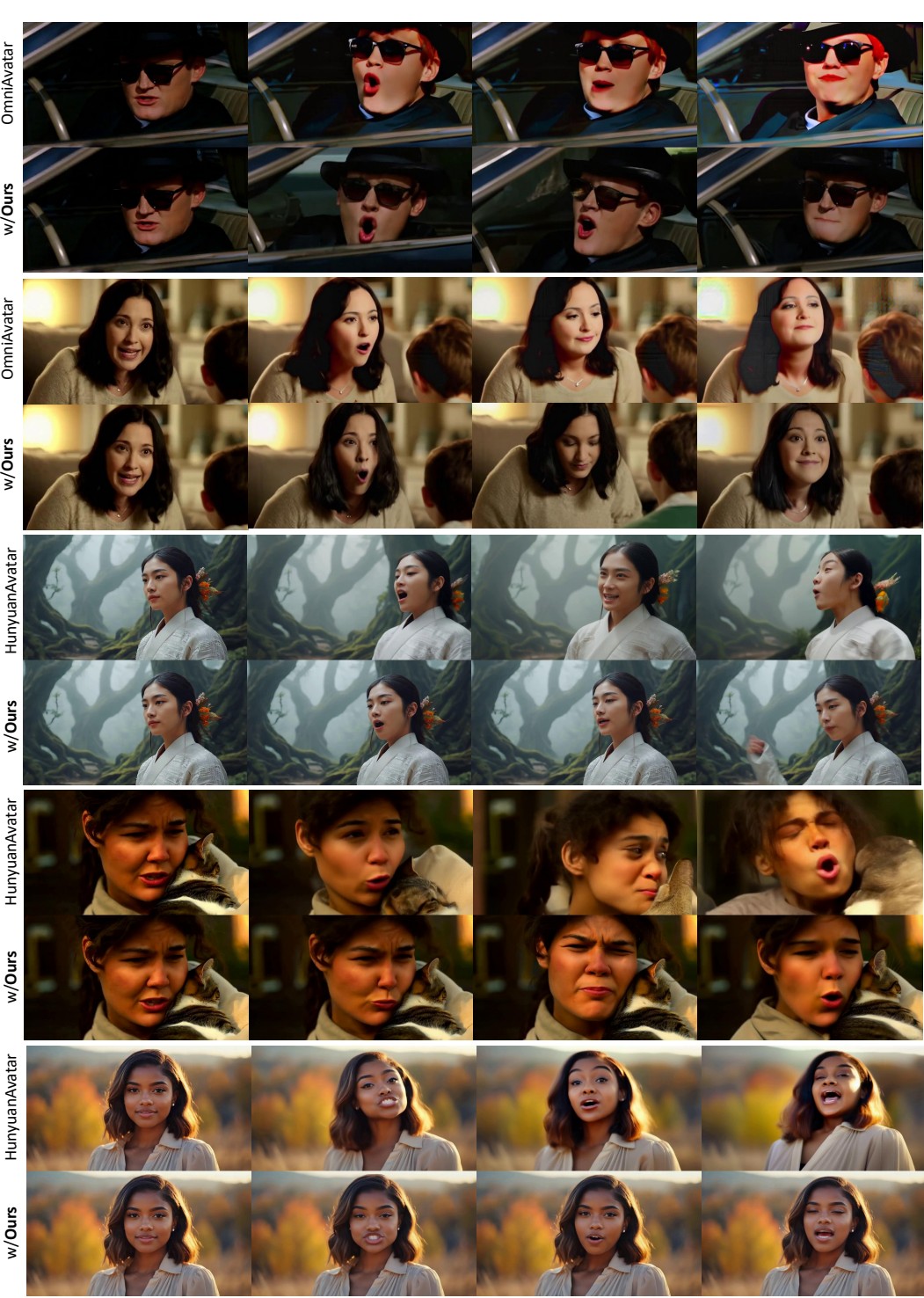

Figure 8: **Additional qualitative results across diverse characters and scenarios.** Our method consistently maintains character identity and visual quality across different subjects, backgrounds, and lighting conditions. Each row compares baseline methods with our Lookahead Anchoring approach, demonstrating superior identity preservation and natural motion generation.

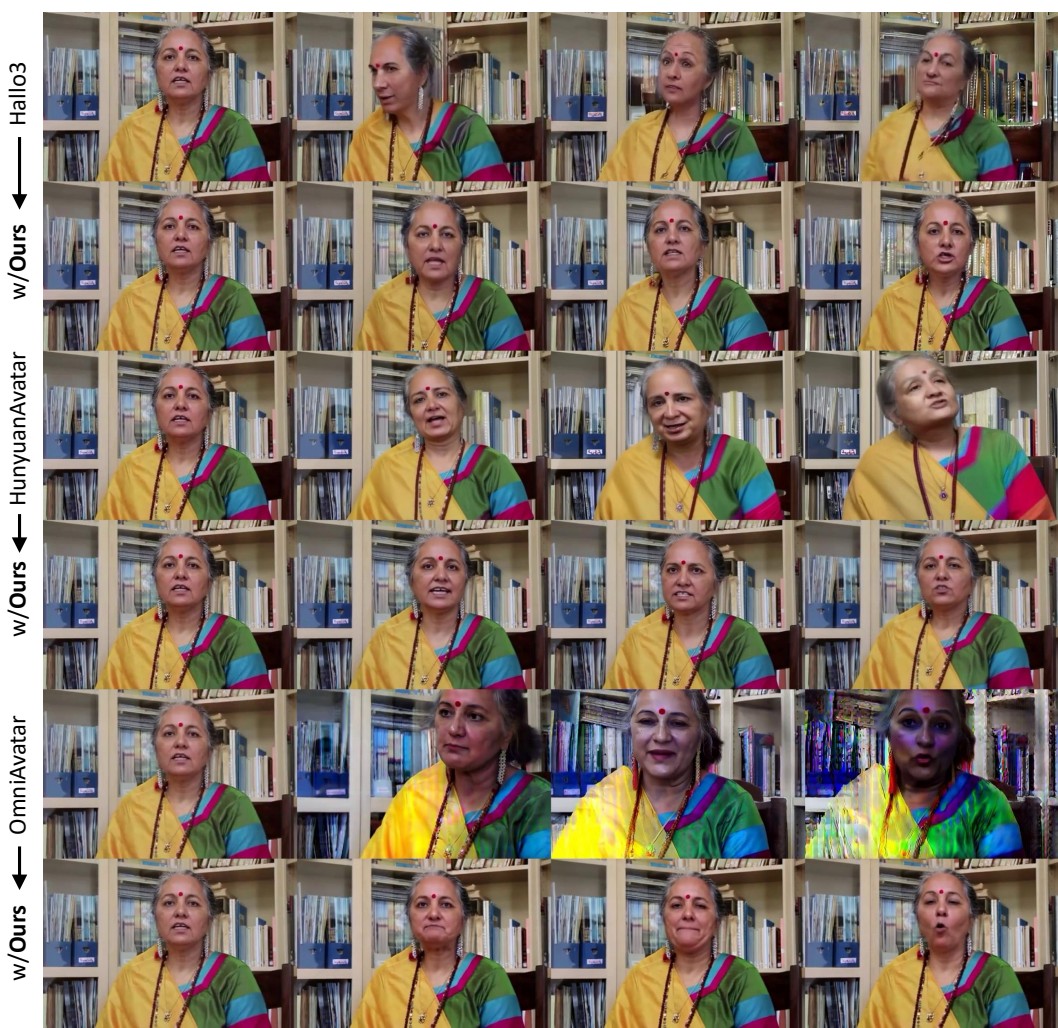

Figure 9: **Additional qualitative comparisons on AVSpeech** (Ephrat et al., 2018). We compare baseline models (top rows) with our Lookahead Anchoring approach (bottom rows). Our method maintains superior character consistency and facial detail preservation throughout extended generation sequences, while baselines exhibit progressive identity drift.

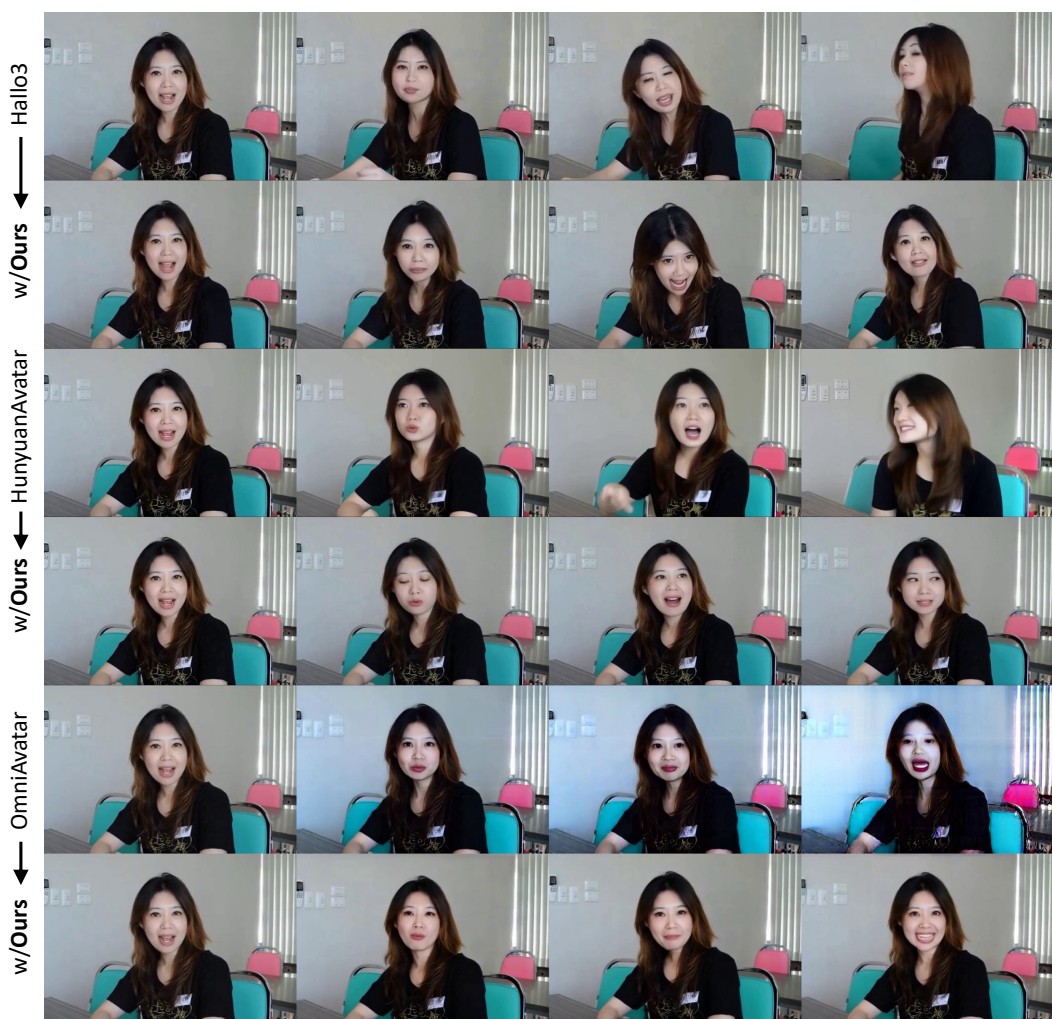

Figure 10: **Additional qualitative comparisons on AVSpeech** (Ephrat et al., 2018). We compare baseline models (top rows) with our Lookahead Anchoring approach (bottom rows). Our method maintains superior character consistency and facial detail preservation throughout extended generation sequences, while baselines exhibit progressive identity drift.

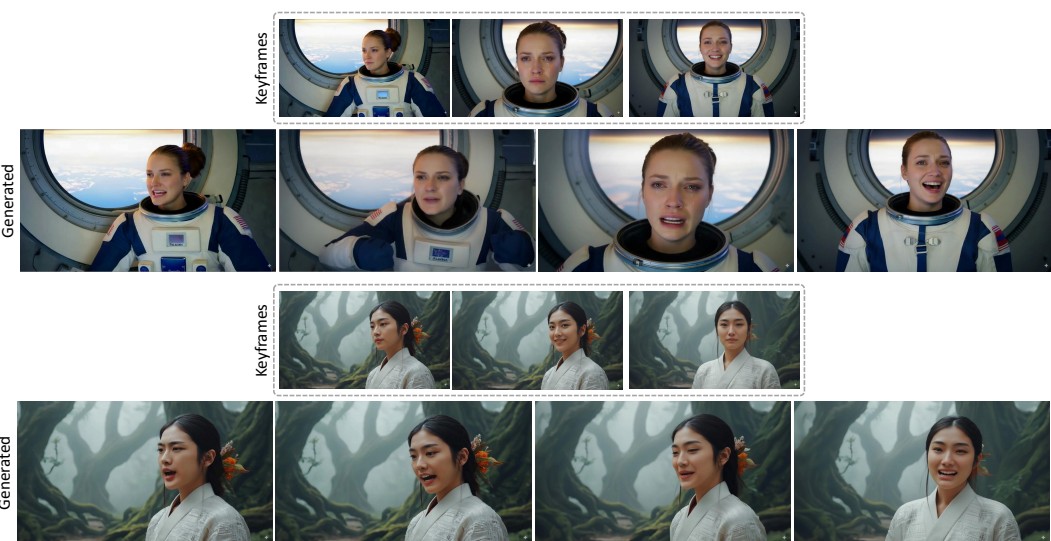

Figure 11: **Narrative-driven long video generation application.** Our approach seamlessly integrates with external text-to-image editing models to create dynamic storylines. By positioning edited reference images as distant lookahead anchors, we achieve smooth narrative transitions while maintaining audio synchronization and character identity throughout the sequence.

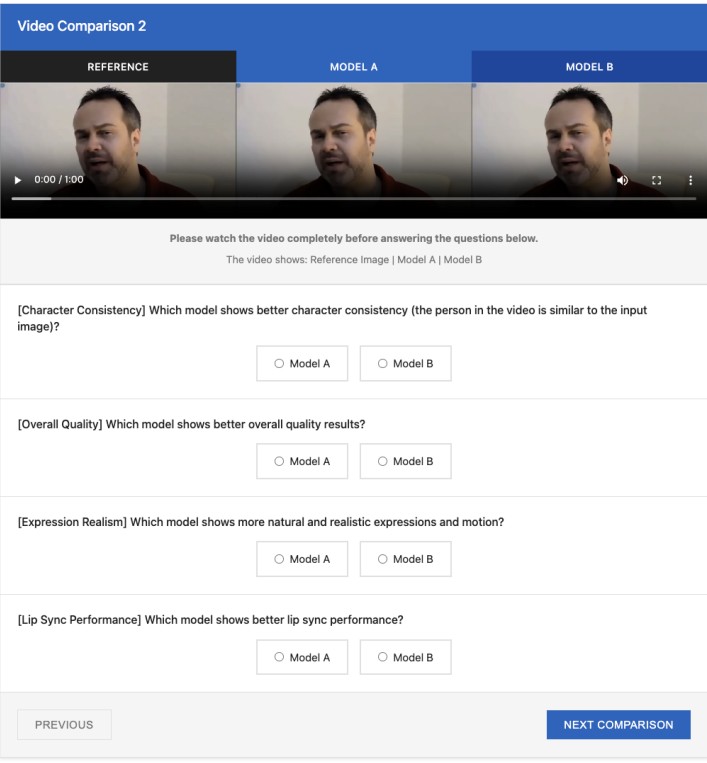

Figure 12: **An example of the screen shown to participants in user study.**

