# OpenReview forum: "Lookahead Anchoring: Preserving Character Identity in Audio-Driven Human Animation"
_ICLR.cc/2026/Conference — Submitted to ICLR 2026_

### Official Review · Reviewer_R6Tg · 2025-10-21

**Soundness:** 3
**Presentation:** 3
**Contribution:** 2
**Rating:** 4
**Confidence:** 5

**Summary:**

This paper finds that the DiT model is limited by the quadratic complexity of the Transformer, allowing it to process only short segments of approximately 5 seconds at a time. To generate longer videos, a segment-based autoregressive generation approach is adopted, but this is prone to character identity drift. New segments rely on previously generated frames, and error accumulation causes the character's appearance to gradually deviate from the original reference image. To address this, the role of keyframes is modified in this paper: keyframes are shifted from being "boundary constraints for the currently generated segment" to "directional guidance for future time steps." While responding to real-time audio signals, the model continuously tracks future keyframes, achieving a balance between identity consistency and motion naturalness.

**Strengths:**

1.	The thinking of sync-free keyframes is reasonable. And the “Do video DiTs understand distant frames” shows how reference frames influence video clip generation clearly.
2.	The long auto-regressive generation results are generally satisfactory, and the motion intensity and identity-preserving abilities are well balanced.
3.	The code and weights will be open sourced, which benefits the reproducing abilities. And it is convincing that several baselines are tested with proposed method.
4.	The writing is clear and easy to understand.

**Weaknesses:**

1.	The proposed distant keyframe conditioning method is not novel enough. Similar methods have been proposed in Section 3.3 of OmniHuman-1.5[1], which are not cited in this paper. As a core part of this work, the originality should be very clear, otherwise it will damage the novelty of this paper.
2.	In the provided demos, the head motions seem to be restricted around a limited area, compared to the baseline methods. I am wondering why this happens. And will the proposed lookahead anchoring restrict the expressiveness ability of motion generation of DiT models, especially in half-body or full-body generation settings?


[1] Jiang J, Zeng W, Zheng Z, et al. Omnihuman-1.5: Instilling an active mind in avatars via cognitive simulation[J]. arXiv preprint arXiv:2508.19209, 2025.

**Questions:**

1.	Discuss the novelty of this paper, compared to OmniHuman-1.5.
2.	Explain the restricted motion shown in demos.
3.	Discuss the proposed method whether could be used to full-body animation setting and the impacts of method.

---

> ### Author Response · Authors · 2025-11-24
>
> We thank the reviewer for finding our thinking on sync-free keyframes reasonable and for confirming that our long generation results are satisfactory with well-balanced motion intensity and identity preservation. We also appreciate the support for our contribution to reproducibility. We address the specific concerns below.
>
> ---
>
> ## Discussion on a concurrent work (OmniHuman1.5).
>
> We thank the reviewer for bringing OmniHuman-1.5 to our attention.
>
> ### [Timeline and Independence]
>
> We respectfully wish to highlight that OmniHuman-1.5 is a concurrent work, released as a non-peer-reviewed preprint only 28 days prior to the ICLR submission deadline. This falls well within the 2-month window defined by ICLR guidelines, under which authors are excused from comparing with such work and being unaware of it.
>
> Our research was conceived and developed independently with a distinct motivation, without knowledge of their approach. In accordance with ICLR guidelines regarding concurrent research, we believe our submission should be evaluated on its independent contributions and merits.
>
> Nevertheless, we agree that discussing similar concurrent work adds value. We have updated our Related Work and Appendix to include this discussion.
>
> ### [Distinctions]
>
> While both methods utilize a future frame to guide generation, the underlying philosophy and training strategies are different:
> * **OmniHuman-1.5 (Inference-time Injection):** This method views the reference image as an "artificial construct" that hinders motion dynamics. Consequently, they **discard the reference image entirely during training**, learning only from native video frames and the reference image is injected only during inference.
> * **Ours (Learnable Temporal Guidance):** In contrast, ours takes a fundamentally different approach by **explicitly incorporating future-sampled image conditions during the training phase.** Our method is designed to enable the model to learn and understand temporal distances between frames, allowing the model to effectively condition on distant keyframes and regulate the influence of context based on temporal proximity, rather than treating it solely as an inference-time technique.
>
> Furthermore, the depth of analysis differs significantly. OmniHuman-1.5's investigation regarding the future frame conditioning is limited to a binary ablation study (validating the module's presence/absence). In contrast, we thoroughly focus on this mechanism and provide comprehensive analyses of the lookahead mechanism through multiple dimensions:
> * Systematic evaluation of lookahead distance as a control parameter (Fig. 6)
> * Exploration of different temporal embedding strategies (Tab. 5)
> * Comparison of fixed versus flexible anchoring training approaches (Tab. 5)
> * Extensive ablations across three distinct architectures demonstrating generalizability.
>
> We have updated our Related Work and Appendix to include this discussion.

---

> > ### Author Response · Authors · 2025-11-24
> >
> > ## Limited motion range of the character.
> >
> > We thank the reviewer for this observation and provide clarification below.
> >
> > ### - Does our method reduce dynamic performance?
> >
> > To address concerns about reduced dynamicity, we quantitatively measured the dynamic degree (following the metric defined in L374) for three base models (Hallo3, HunyuanAvatar, OmniAvatar) both with and without our method.
> >
> > | | Baseline | +Ours | Real Videos (GT) |
> > | :--- | :--- | :--- | :--- |
> > | **Hallo3** | 0.6759 | 0.8784 | 0.4328 |
> > | **HunyuanAvatar** | 0.7069 | 0.4779 | 0.4328 |
> > | **OmniAvatar** | 0.2755 | 0.4561 | 0.4328 |
> >
> > The results demonstrate that our method's impact on motion dynamics is model-dependent rather than universally restrictive. Notably, dynamicity increases for Hallo3 and OmniAvatar while decreasing for HunyuanAvatar. This pattern suggests our method adapts to each model's specific degradation characteristics: HunyuanAvatar tends to accumulate errors that manifest as excessive and unnatural motion, while OmniAvatar becomes overly static.
> >
> > The Real Video (GT) baseline of 0.4328 provides context for interpreting these metrics. The varying responses across models indicate that our method provides adaptive rather than uniform motion regulation.
> >
> >
> > ### - Prompt-Dependent Motion Generation
> >
> > We also clarify that the relatively monotonous upper-body and hand motion in some results stems primarily from the neutral prompt (“a person is speaking”). We observed our method generates more dynamic upper-body and hand movements when explicitly prompted. To demonstrate this, we have included a new result in the revised Supplementary Video. This video, generated with the prompt “A person is speaking with animated gestures” (based on OmniAvatar+Ours), exhibits more expressive dynamics. We will include this analysis and additional high-dynamic samples in the final version.
> >
> > ---
> >
> > ## Full-body animation.
> >
> > We thank the reviewer for the constructive suggestion. We have included full-body animation results in the revised Supplementary Video to demonstrate the model's capability in generating whole-body dynamics.

---

> ### Comment · Reviewer_R6Tg · 2025-11-26
>
> I don't quite understand the precise meaning of your statement, "provides adaptive rather than uniform motion regulation". From the results in the table you provided, the Hallo3-Baseline has higher dynamics than the ground truth (GT), but the proposed method still increases the dynamics. However, the HunyuanAvatar-Baseline also has higher dynamics than the GT, while the proposed method reduces the dynamics in this case. I haven't been able to summarize the pattern of how the proposed method affects the dynamics of the baseline method.
>
> Furthermore, I don't believe the "neutral prompt" issue is a significant reason for the dynamics changes, because you should have used the same prompt for the baseline and +ours tests.
>
> It would be better if you could discuss these points further.

---

> ### Author Response · Authors · 2025-11-26
>
> We apologize for the confusion in our previous response and appreciate the opportunity to clarify!
>
> We intended to clarify that the changes in dynamics are model-dependent consequences of maintaining identity consistency, rather than a uniform reduction.
>
> Across all cases, dynamics with our method remain higher than GT. Under the assumption that dynamics closer to GT is desirable, our method improves this metric for some models (HunyuanAvatar, OmniAvatar) while increasing the deviation for others (Hallo3). Therefore, our method does not universally reduce dynamics but rather affects them differently based on the base models.
>
> Regarding the prompt issue, we agree with the reviewer that this does not explain the dynamics difference between baseline and ours. As shown in Fig. 6 (on HunyuanAvatar), there is indeed a trade-off between identity preservation and dynamics. Our point regarding prompting was to demonstrate that the restricted motion observed in the some demos can be mitigated through orthogonal means (e.g., expressive prompts), independent of our identity preservation approach.
>
> We appreciate the reviewer's active feedback and are glad to address these concerns. We will incorporate these clarifications in the revision. Please feel free to raise any remaining questions.

---

### Official Review · Reviewer_DiP9 · 2025-10-30

**Soundness:** 2
**Presentation:** 2
**Contribution:** 1
**Rating:** 2
**Confidence:** 4

**Summary:**

This paper proposes Lookahead Anchoring approach to address the problem of identity drift in long-form, audio-driven human animation.
Instead of forcing the model to meet specific keyframes at segment boundaries, keyframes are placed at a future time step (ahead of the current generation window), pushing the model to "chase" them. Such a design enables the preservation of character identity while allowing for more expressive motion dynamics. Experiments show that by using the Lookahead Anchoring strategy, the model can  maintain consistent identity over time while generating plausible audio-driven human animation.

**Strengths:**

* The method is demonstrated to generalize across multiple DiT-based human animation models including Hallo3, OmniAvatar, HunyuanAvatar (Sec. 4.1), which showcases its broad applicability and the potential for integration into other architectures.

* The paper presents both quantitative and qualitative results showing the superiority of the proposed approach. In experiments with long video generation, Lookahead Anchoring outperforms traditional methods in terms of character consistency and overall video quality.

**Weaknesses:**

* The concept of Lookahead Anchoring is not a new thing. Similar ideas have been explored in prior works like Omnihuman-1.5 (released on arXiv one month before the ICLR paper deadline), which also introduces a  Pseudo Last Frame design to anchor the given reference frame  at future timesteps ahead of the current generation window. Unfortunately, the paper does not cite or discuss these existing methods.

* The results in the supplemental video (02:56-04:35) suggest that the Lookahead Anchoring strategy limits the motion range of the character. This restriction may hinder the model's ability to generate highly dynamic and expressive animations, which could be a significant drawback for certain use cases requiring more fluid motion.

* Even if the Lookahead Anchoring strategy is completely novel, it is relatively simple and may not be novel enough to support a paper at ICLR, which typically expects more advanced and intricate contributions. The simplicity of the method, though effective for certain scenarios, may not meet the high standards of innovation and complexity expected at this level of the conference.

* While the paper does a good job focusing on identity preservation and lip synchronization for simple scenarios, more experiments on scene dynamics, such as handling large environmental changes (e.g, view changes or moving background) are necessary. Current discussion on these cases is relatively brief.

**Questions:**

See [Weaknesses]

---

> ### Author Response · Authors · 2025-11-24
>
> We thank the reviewer for validating the broad applicability of our method across multiple DiT-based models. We also appreciate the acknowledgment of our approach's superiority over traditional methods, particularly in terms of character consistency and overall video quality. We address the specific concerns below.
>
> ---
>
> ## The simplicity of the method may not meet the high standards of innovation and complexity expected at this level of the conference.
>
> We respectfully disagree with the premise that complexity is a prerequisite for contribution.
>
> We believe impactful advances often arise from straightforward insights that prove broadly applicable. Our method's simplicity enables seamless integration with diverse DiT-based models (Hallo3, HunyuanAvatar, OmniAvatar) without architectural modifications, establishing it as a generalized, simple yet effective solution to address identity drift in long video generation.
>
> Our work reveals a latent intrinsic capability of Video DiTs to understand distant temporal relationships (Sec 3.3). Lookahead Anchoring is a mechanism that leverages this property to solve identity drift, not merely an engineering heuristic.
>
> The proposed simple yet effective method is backed by rigorous analysis; we identify "lookahead distance" as a control mechanism for the expressiveness-identity trade-off and introduce a specialized flexible anchoring strategy that goes beyond naive fine-tuning to maximize performance (Sec 4.3).
>
>
> ---
>
> ## Discussion on a concurrent work (OmniHuman1.5).
>
> We thank the reviewer for bringing OmniHuman-1.5 to our attention.
>
> ### [Timeline and Independence]
>
> We respectfully wish to highlight that OmniHuman-1.5 is a concurrent work, released as a non-peer-reviewed preprint only 28 days prior to the ICLR submission deadline. This falls well within the 2-month window defined by ICLR guidelines, under which authors are excused from comparing with such work and being unaware of it.
>
> Our research was conceived and developed independently with a distinct motivation, without knowledge of their approach. In accordance with ICLR guidelines regarding concurrent research, we believe our submission should be evaluated on its independent contributions and merits.
>
>
> ### [Distinctions]
>
> While both methods utilize a future frame to guide generation, the underlying philosophy and training strategies are different:
> * **OmniHuman-1.5 (Inference-time Injection):** This method views the reference image as an "artificial construct" that hinders motion dynamics. Consequently, they **discard the reference image entirely during training**, learning only from native video frames and the reference image is injected only during inference.
> * **Ours (Learnable Temporal Guidance):** In contrast, ours takes a fundamentally different approach by **explicitly incorporating future-sampled image conditions during the training phase.** Our method is designed to enable the model to learn and understand temporal distances between frames, allowing the model to effectively condition on distant keyframes and regulate the influence of context based on temporal proximity, rather than treating it solely as an inference-time technique.
>
> Furthermore, the depth of analysis differs significantly. OmniHuman-1.5's investigation regarding the future frame conditioning is limited to a binary ablation study (validating the module's presence/absence). In contrast, we thoroughly focus on this mechanism and provide comprehensive analyses of the lookahead mechanism through multiple dimensions:
> * Systematic evaluation of lookahead distance as a control parameter (Fig. 6)
> * Exploration of different temporal embedding strategies (Tab. 5)
> * Comparison of fixed versus flexible anchoring training approaches (Tab. 5)
> * Extensive ablations across three distinct architectures demonstrating generalizability.
>
> We have updated our Related Work and Appendix to include this discussion.

---

> ### Author Response · Authors · 2025-11-24
>
> ## Limited motion range of the character.
>
> We thank the reviewer for this observation and provide clarification below.
>
> ### - Does our method reduce dynamic performance?
>
> To address concerns about reduced dynamicity, we quantitatively measured the dynamic degree (following the metric defined in L374) for three base models (Hallo3, HunyuanAvatar, OmniAvatar) both with and without our method.
>
> | | Baseline | +Ours | Real Videos (GT) |
> | :--- | :--- | :--- | :--- |
> | **Hallo3** | 0.6759 | 0.8784 | 0.4328 |
> | **HunyuanAvatar** | 0.7069 | 0.4779 | 0.4328 |
> | **OmniAvatar** | 0.2755 | 0.4561 | 0.4328 |
>
> The results demonstrate that our method's impact on motion dynamics is model-dependent rather than universally restrictive. Notably, dynamicity increases for Hallo3 and OmniAvatar while decreasing for HunyuanAvatar. This pattern suggests our method adapts to each model's specific degradation characteristics: HunyuanAvatar tends to accumulate errors that manifest as excessive and unnatural motion, while OmniAvatar becomes overly static. The Real Video (GT) baseline of 0.4328 provides context for interpreting these metrics.
>
>
> ### - Prompt-Dependent Motion Generation
>
> We also clarify that the monotonous upper-body and hand motion in some results stems primarily from the neutral prompt (“a person is speaking”). We observed our method generates more dynamic upper-body and hand movements when explicitly prompted. To demonstrate this, we have included a new result in the revised Supplementary Video. This video, generated with the prompt “A person is speaking with animated gestures” (based on OmniAvatar+Ours), exhibits more expressive dynamics. We will include this analysis and additional high-dynamic samples in the final version.
>
> ---
>
> ## Discussion on Dynamic Background.
>
> We appreciate the feedback. Since our contribution focuses on preventing identity drift via a lightweight adaptation, our method does not introduce separate background modeling but rather preserves the generative priors of the pre-trained backbones. Consequently, the handling of dynamic backgrounds depends on the base model's original capabilities. We have added examples to the Supplementary Video (based on OmniAvatar) demonstrating that our method retains the base model's ability to generate non-static backgrounds. We will incorporate this discussion into the final version.

---

### Official Review · Reviewer_aq1t · 2025-10-31

**Soundness:** 4
**Presentation:** 3
**Contribution:** 2
**Rating:** 4
**Confidence:** 5

**Summary:**

Audio-driven human animation suffers from identity drift in long videos. Existing fixes either need extra keyframe models or restrict natural motion, failing to balance identity consistency and motion freedom.

This paper proposes Lookahead Anchoring: using future-timestamp keyframes as "guides" instead of current-window ones. It uses the reference image directly as the future anchor and adjusts lookahead distance to balance identity and motion.

Tests on three DiT-based models and datasets show it boosts identity consistency, maintains lip synchronization, and improves video quality. It also supports narrative-driven generation, serving as a practical solution for long audio-driven animations.

**Strengths:**

1. This paper proposes a new keyframe logic, which differs from traditional methods like KeyFace that rely on rigid boundary constraints or other reference-net-based designs. It converts keyframes into future-oriented guides, named self-keyframing, aiming to maintain character identity and address error accumulation.
2. The approach designs temporal distance as a controllable parameter: smaller D values prioritize identity adherence, larger D values focus on motion expressivity.
3. The method is integrated with three DiT-based audio-driven models (Hallo3, HunyuanVideo-Avatar, OmniAvatar) through a fine-tuning strategy. This integration is meant to show that the method can be applied to multiple architectures, not just a custom model framework.
4. The work explores narrative-driven long video generation by combining text-based image editing models to create story-specific keyframes, thereby enhancing the solution’s extensibility to meet varied scenario-based requirements.

**Weaknesses:**

1. The method mentioned in Section 3.3 of the OmniHuman1.5[1] is almost identical to this work, so I have some doubts about the innovativeness—nevertheless, this work features more detailed experiments compared to that paper.
2. The method's visualizations do demonstrate its capability in generating long-duration videos, yet it lacks performance in high-dynamic scenarios: character dynamics remain relatively monotonous, with limited upper-body and hand movements.
3. It would be good to visualize the ablation study for the distant keyframe conditioning.

[1] Jianwen Jiang, Weihong Zeng, Zerong Zheng and et.al. OmniHuman-1.5: Instilling an Active Mind in Avatars via Cognitive Simulation

**Questions:**

1. The first part of the qualitative comparison section in the supplementary materials (featuring a woman with wavy curly hair in a blue outfit), the dynamic effect of the result with lookahead anchoring is weaker than that of the baseline without it. This raises the question of whether this method might reduce dynamic performance？
2. The paper mentions that the strategy of directly using anchors without training will produce artifacts. If you directly discard the final latent with artifacts in longer video generation tasks, can this serve as a training-free method?
3. Three different baselines are used in this paper, which all perform well in fixed scenarios. However, have you tried using models with camera movement capabilities to verify the effectiveness of this method? And could the anchor frame possibly restrict the range of camera movement?

---

> ### Author Response · Authors · 2025-11-24
>
> We thank the reviewer for highlighting our design of temporal distance as a controllable parameter to balance identity and expressivity. We are also encouraged by the recognition of our method's integration with multiple architectures and its extensibility to narrative-driven scenarios. We address the specific concerns below.
>
> ---
>
> ## Discussion on a concurrent work (OmniHuman1.5).
>
> We thank the reviewer for bringing OmniHuman-1.5 to our attention and for acknowledging the more detailed experiments in our work.
>
> ### [Timeline and Independence]
>
> We respectfully wish to highlight that OmniHuman-1.5 is a **concurrent work**, released publicly only 28 days prior to the ICLR submission deadline. Our research was conceived and developed independently with a distinct motivation, without knowledge of their approach. In accordance with ICLR guidelines regarding concurrent research, we believe our submission should be evaluated on its independent contributions and merits.
>
> ### [Distinctions]
>
> While both methods utilize a future frame to guide generation, the underlying philosophy and training strategies are different:
> * **OmniHuman-1.5 (Inference-time Injection):** This method views the reference image as an "artificial construct" that hinders motion dynamics. Consequently, they **discard the reference image entirely during training**, learning only from native video frames and the reference image is injected only during inference.
> * **Ours (Learnable Temporal Guidance):** In contrast, ours takes a fundamentally different approach by **explicitly incorporating future-sampled image conditions during the training phase.** Our method is designed to enable the model to learn and understand temporal distances between frames, allowing the model to effectively condition on distant keyframes and regulate the influence of context based on temporal proximity, rather than treating it solely as an inference-time technique.
>
> Furthermore, the depth of analysis differs significantly as the reviewer highlighted. OmniHuman-1.5's investigation regarding the future frame conditioning is limited to a binary ablation study (validating the module's presence/absence). In contrast, we thoroughly focus on this mechanism and provide comprehensive analyses of the lookahead mechanism through multiple dimensions:
> * Systematic evaluation of lookahead distance as a control parameter (Fig. 6)
> * Exploration of different temporal embedding strategies (Tab. 5)
> * Comparison of fixed versus flexible anchoring training approaches (Tab. 5)
> * Extensive ablations across three distinct architectures demonstrating generalizability.
>
> We have updated our Related Work and Appendix to include this discussion.
>
> ---
>
> ## Motion dynamicity and highly dynamic scenarios.
>
> We thank the reviewer for this observation and provide clarification below.
>
> ### - Does our method reduce dynamic performance?
>
> To address concerns about reduced dynamicity, we quantitatively measured the dynamic degree (following the metric defined in L374) for three base models (Hallo3, HunyuanAvatar, OmniAvatar) both with and without our method.
>
> | | Baseline | +Ours | Real Videos (GT) |
> | :--- | :--- | :--- | :--- |
> | **Hallo3** | 0.6759 | 0.8784 | 0.4328 |
> | **HunyuanAvatar** | 0.7069 | 0.4779 | 0.4328 |
> | **OmniAvatar** | 0.2755 | 0.4561 | 0.4328 |
>
> The results demonstrate that our method's impact on motion dynamics is model-dependent rather than universally restrictive. Notably, dynamicity increases for Hallo3 and OmniAvatar while decreasing for HunyuanAvatar. This pattern suggests our method adapts to each model's specific degradation characteristics: HunyuanAvatar tends to accumulate errors that manifest as excessive and unnatural motion, while OmniAvatar becomes overly static. The Real Video (GT) baseline of 0.4328 provides context for interpreting these metrics.
>
>
> ### - Limited upper-body and hand movements
>
> We clarify that the monotonous upper-body and hand motion in some results stems primarily from the neutral prompt (“a person is speaking”). We observed our method generates more dynamic upper-body and hand movements when explicitly prompted. To demonstrate this, we have included a new result in the revised Supplementary Video. This video, generated with the prompt “A person is speaking with animated gestures” (based on OmniAvatar+Ours), exhibits more expressive dynamics. We will include this analysis and additional high-dynamic samples in the final version.

---

> > ### Author Response · Authors · 2025-11-24
> >
> > ## If you directly discard the final latent with artifacts in longer video generation tasks, can this serve as a training-free method?
> >
> > We thank the reviewer for this insightful question regarding the feasibility of a training-free approach.
> >
> > First, we wish to clarify that our proposed method already discards the future frame latents used as conditions during inference, with the exception of the 2-frame pilot experiment shown in Fig. 3.
> >
> > To empirically verify if our method serves as a training-free method, we conducted an experiment using OmniAvatar as the base model during the rebuttal period:
> >
> > | | Sync-D | Face Consistency | Dynamic Degree |
> > | :--- | :--- | :--- | :--- |
> > | **Training-free variant** | 7.89 | 0.673 | 0.343 |
> > | **Ours** | **7.30** | **0.780** | **0.563** |
> >
> > As shown in the table, the training-free variant exhibits only marginal performance compared to our full method.
> >
> > ---
> >
> > ## Ours with camera-controllable video generative models.
> >
> > We thank the reviewer for this constructive suggestion. Our work has focused specifically on human-oriented video generation with relatively static viewpoints typical of talking head scenarios. While camera controllability was not within our primary scope, we recognize its importance for broader applicability.
> >
> > Given that our method allows the model to decouple from immediate reference poses at larger lookahead distances, we hypothesize it could accommodate flexible camera trajectories while preserving consistency.
> >
> > Due to computational constraints during the rebuttal period, we could not complete full fine-tuning experiments on camera-controllable baselines. We acknowledge this as an important direction and will include results and discussion in the camera-ready version.
> >
> >
> > ---
> >
> > ## Visualization of the ablation study for the distant keyframe conditioning.
> >
> > We appreciate this suggestion and have added visualizations of this ablation to the **revised Supplementary Video**. The final version will include additional diverse video results that could not be included in the current revision due to file size limitations.

---

### Official Review · Reviewer_fgaf · 2025-10-31

**Soundness:** 4
**Presentation:** 4
**Contribution:** 3
**Rating:** 6
**Confidence:** 4

**Summary:**

This paper proposes Lookahead Anchoring (LA), a method designed to preserve character identity in audio-driven human animation. Existing methods rely on keyframe generation producing intermediate frames  and identify specific feature inject to prevent identity drift. But these explicit keyframes can overly constrain the motion dynamics, limiting natural expressivity.

To overcome this, LA introduces future keyframe conditioning: rather than generating anchors within the current sequence, the model leverages future latent keyframes as soft temporal guidance.

Key observations include:
	•	The temporal lookahead distance directly balances expressivity and consistency; larger distances produce more dynamic motion, smaller ones yield stronger identity adherence.
	•	LA integrates seamlessly into existing DiT-based architectures and improves both identity stability and lip synchronization across long sequences.

**Strengths:**

1. Conceptual innovation: The idea of using future latent frames as temporal anchors instead of rigidly generated keyframes is elegant and conceptually clear. It shifts the paradigm from hard constraints to soft temporal guidance.
2. Interpretability: The empirical finding that lookahead distance controls a trade-off between motion expressivity and identity consistency is intuitive and well-supported (Fig. 6).
3. Model-agnostic integration: LA can be attached to existing transformer or diffusion-based animation models with minimal architectural change.
4. Quantitative gains: Across HDTF and AVSpeech datasets, LA consistently improves lip synchronization (Sync-D ↓, Sync-C ↑), face/subject consistency, and perceptual quality (FID ↓, FVD ↓), without harming motion smoothness.
5. Perceptual preference: User studies show strong preference for LA-enhanced videos in terms of synchronization and identity stability.
6. Practical significance: The approach removes the need for an explicit keyframe generation stage, simplifying pipelines for identity-preserving video generation.

**Weaknesses:**

* Missing justification for “bounded generation” argument
The introduction claims that “bounded” keyframe-based methods are limited by the quality and expressiveness of their generated keyframes. While this is plausible, the paper does not provide quantitative or visual evidence demonstrating this limitation.

* Ambiguity in “self-keyframing” explanation
The statement that “the keyframe no longer needs to match the exact lip movements and expressions required by the audio … enabling self-keyframing” is conceptually interesting but under-explained. It’s unclear how a distant or reference-based anchor can substitute for synchronised keyframes in guiding expression or pose accuracy.

* Lack of comparison with KeyFace (Bigata et al., 2025)
KeyFace is currently a strong state-of-the-art method for identity-preserving talking heads, explicitly designed to address identity degradation. Its absence from the qualitative comparisons leaves a significant gap. A side-by-side video comparison would substantially strengthen the evaluation.

*  Limited clarity on latent-space interpretation
As I understand it, each latent token represents a spatiotemporally compressed patch, not a full frame. Appending the lookahead latent z_{n-1+d} therefore adds only one additional patch-level token, not a holistic future-frame reference. It is unclear how this single patch provides global temporal guidance or identity stabilization across the entire sequence.

* Inference-time mechanism under-specified
During inference, when only one keyframe or reference image is available, it remains unclear how the lookahead mechanism functions in practice. Does the model still benefit from a meaningful anchor signal? If not, this could contradict the paper’s claim that LA allows “sync-free keyframes” capable of matching poses and expressions.

**Questions:**

1.	Could you clarify what “bounded generation” refers to in practice, and show evidence that conventional keyframe methods limit expressiveness or quality?
2.	How exactly does “self-keyframing” function; does the model reuse its own generated frames, or the original reference frame, as recursive anchors?
3.	Why was KeyFace excluded from qualitative comparisons, given its strong relevance to identity preservation?
4.	Given that each latent token represents a spatiotemporal patch, how can a single appended future latent provide meaningful global identity anchoring?
5.	During inference with only one keyframe/reference image, how is the lookahead mechanism applied, and does it still contribute to identity consistency?

---

> ### Author Response · Authors · 2025-11-24
>
> We thank the reviewer for recognizing our approach as elegant and conceptually clear. We also appreciate the positive remarks on the intuitive trade-off regarding lookahead distance and our consistent quantitative gains. We address the specific concerns below.
>
> ---
>
> ## Clarification for "Bounded Generation” and comparison with KeyFace.
>
> "Bounded generation" refers to a structural limitation where the final video quality is strictly capped by the quality of explicitly generated keyframes. Methods like KeyFace enforce hard constraints at these steps, often resulting in reduced realism and limited expressivity compared to the soft guidance provided by our approach.
>
> To demonstrate this visually as suggested, we **have included a side-by-side comparison with KeyFace in the revised Supplementary Video**. This qualitative comparison highlights the limited expressiveness of the baseline, which aligns with the quantitative results presented in Tab.2 (KeyFace) and Tab. 4 (keyframe-based baseline with the same model).
>
> ---
>
> ## Inference-time mechanism when only one keyframe is available.
>
> We appreciate the reviewer's question. Our approach handles this scenario through two mechanisms described in the paper:
>
> **Self-keyframing (L223, Tables 1-2):** During inference with a single reference image, we inject the reference identity image (encoded as latents) into the lookahead position as recursive anchors, as detailed in L223. Results in Table 1 and Table 2 were generated using this protocol. This approach is effective because the model is trained to utilize the lookahead anchor primarily for identity consistency rather than pose alignment. Unlike existing methods that require the keyframe to perfectly match the target timestamp, our model learns to extract high-level identity features from the distant anchor.
>
> Consequently, even though the static single reference image does not match the target pose or lip shape of the future frame, it provides a robust anchor signal that maintains identity throughout the sequence without interfering with the audio-driven motion.
>
> **Our method with multiple frames (L225, Fig.7):** We can also leverage image generation models to create narrative-appropriate keyframes, which then serve as anchors. Our method can utilize multiple keyframes as in conventional keyframe-based methods, and these keyframes can be either generated through image generation models or provided by users.
>
> Both approaches provide meaningful anchor signals for identity preservation. The key insight is that our lookahead mechanism naturally decouples identity preservation from pose/expression matching when the temporal distance increases, enabling robust single-image inference without the limitations of methods that require frame-accurate keyframes.
>
>
>
> ---
>
> ## Latent token.
>
> We clarify that the variable $z_n$ in our paper denotes the complete set of latent tokens corresponding to  frame $n$ (i.e., the flattened sequence of tokens representing the entire spatial dimension). Therefore, appending the lookahead latent $z_{n-1+d}$ provides holistic, global spatial context for the future frame, rather than just a local patch, enabling the effective identity anchoring and temporal stability observed in our results.
>
> We have revised Section 3 to define the latent notation more explicitly, and hope this has addressed the reviewer’s concern.

---

### Meta-Review · Area_Chair_tynU · 2026-01-02

**Summary:**

While it is acknowledged that the writing is clear and consistent empirical gains are achieved when the method is applied to multiple existing DiT-based audio-driven animation backbones, the core idea of repositioning keyframes as temporally distant “lookahead” anchors was viewed as a simple heuristic and not clearly establishing a new principle or algorithmic advance. Several reviewers raised concerns about motion expressiveness and dynamics, suggesting the approach may trade off motion range or lack convincing evidence of robustness beyond the evaluated setting.

**Reviewer Concerns:**

Addressed: 1) comparison with KeyFace: authors state they added side-by-side comparisons with KeyFace in the revised table and supplementary video; 2)latent-token misunderstanding: authors clarified their notation corresponds to the full set of spatial tokens for a frame, and it is not a single patch token; 3) training-free variant: a new comparison showing the training-free variant is weaker than their full training strategy, which indicates the need for training.

Not fully addressed: 1) motion dynamics/expressiveness trade-off: authors provided a “dynamic degree” table and argued effects are model-dependent rather than uniformly restrictive; however, it is not clear when and why motion becomes restricted in practice, and whether the behavior is predictable/controllable beyond tuning a heuristic distance parameter. 2) The core contribution is limited and heuristic. The rebuttal argues simplicity as a virtue but does not provide a stronger technical framing, such as a principled objective or a broadly applicable mechanism beyond a temporal offset anchor.

**Reviewer Scores:**

The reviewers haven't shown any intention to change their initial ratings, given the concerns summarized above.

---

### Decision · Program_Chairs · 2026-01-26

Reject